# Acute Fluid Intake Impacts Assessment of Body Composition via Bioelectrical Impedance Analysis. A Randomized, Controlled Crossover Pilot Trial

**DOI:** 10.3390/metabo13040473

**Published:** 2023-03-26

**Authors:** Janis Schierbauer, Svenja Günther, Sandra Haupt, Rebecca T. Zimmer, Daniel Herz, Thomas Voit, Paul Zimmermann, Nadine B. Wachsmuth, Felix Aberer, Othmar Moser

**Affiliations:** 1Division of Exercise Physiology and Metabolism, Department of Sport Science, University of Bayreuth, 95440 Bayreuth, Germany; svenja.guenther@uni-bayreuth.de (S.G.); sandra.haupt@uni-bayreuth.de (S.H.); rebecca.zimmer@uni-bayreuth.de (R.T.Z.); daniel.herz@uni-bayreuth.de (D.H.); thomas.voit@uni-bayreuth.de (T.V.); paul.zimmermann@uni-bayreuth.de (P.Z.); nadine.wachsmuth@uni-bayreuth.de (N.B.W.); felix.aberer@medunigraz.at (F.A.);; 2Interdisciplinary Metabolic Medicine Trials Unit, Division of Endocrinology and Diabetology, Department of Internal Medicine, Medical University of Graz, 8036 Graz, Austria

**Keywords:** sodium chloride, Ringer, glucose, electrolytes, intracellular water, extracellular water, skeletal muscle mass, fat mass, visceral fat

## Abstract

Bioelectrical impedance analysis (BIA) has proven to be particularly useful due to its inexpensive and rapid assessment of total body water and body density. However, recent fluid intake may confound BIA results since equilibration of fluid between intra- and extracellular spaces may take several hours and furthermore, ingested fluids may not be fully absorbed. Therefore, we aimed to evaluate the impact of different fluid compositions on the BIA. A total of eighteen healthy individuals (10 females, mean ± SD age of 23.1 ± 1.8 years) performed a baseline measurement of body composition before they consumed isotonic 0.9% sodium-chloride (ISO), 5% glucose (GLU) or Ringer (RIN) solutions. During the visit of the control arm (CON), no fluid was consumed. Further impedance analyses were conducted every 10 min after the fluid consumption for 120 min. We found statistically significant interactions between the effects of solution ingestion and time for intra- (ICW, *p* < 0.01) and extracellular water (ECW, *p* < 0.0001), skeletal muscle mass (SMM, *p* < 0.001) and body fat mass (FM, *p* < 0.01), respectively. Simple main effects analysis showed that time had a statistically significant effect on changes in ICW (*p* < 0.01), ECW (*p* < 0.01), SMM (*p* < 0.01) and FM (*p* < 0.01), while fluid intake did not have a significant effect. Our results highlight the importance of a standardized pre-measurement nutrition, with particular attention to hydration status when using a BIA for the evaluation of body composition.

## 1. Introduction

Body composition analysis is a relevant tool to assess the hydration and nutritional status in both health and diseases [1,2,3]. A common method to assess body composition is by means of a bioelectrical impedance analysis (BIA). BIA is a non-invasive technique that estimates body composition based on the measurement of an electrical impedance. It has gained popularity due to its quick, easy and inexpensive application, becoming a valuable tool in clinical, scientific, and fitness settings [3,4]. Especially in the clinical setting, impedance measurements play a vital role in numerous areas, including electrical impedance tomography [5,6], impedance myography [7,8] or surface electromyography [9,10]. BIA measures the resistance of body tissues to an electrical current, sending an electrical signal through the body. The physiology behind this calculation is the varying resistance of different types of tissues, including body water, fat, muscle, and bone, which is proportional to their water content [11,12]. Therefore, the results obtained by the BIA are calculated especially based on the distribution of total body water, i.e., intra- and extracellular water. Next to several factors that can influence the performance of BIA results, e.g., age, sex or body fat distribution, the magnitude of total body water (TBW) content especially is one of the most significant factors [4,11,13]. Total body water directly affects the conductivity of the electrical current, which is why dehydration can lead to an overestimation of body fat, while overhydration can result in an underestimation. Acute changes in TBW via fluid intake have been shown to increase the electrical resistance of tissues; however, the time course of these changes is initially dependent on the ionic composition of the fluids [14]. This implies that in addition to the volume expansion alone, the electrolyte content of the respective fluids also plays a major role in affecting the electrical resistance during a BIA. Previous studies on the changes in body composition administering iso-, hypo and/or hypertonic fluids have led to inconsistent findings [15,16,17,18], which are mainly explained by confounding effects such as uncontrolled exercise, heat exposure, sweat loss or an insufficient number of measurements after the fluid ingestion. The latter is of particular importance because fluid absorption may take up to 1–2 h [19], which is especially the case after the consumption of large fluid volumes.

To date, only few investigations have addressed in detail the acute effects of volume expansion via orally administered fluids containing osmotically active components such as sodium and chloride, and a less osmotically active component such as glucose. Additionally, taking fluid absorption into account would allow for a more precise statement as to whether volume expansion alone or combined with electrolyte administration is more likely to influence changes in intra- and extracellular water and thus body composition.

Therefore, the aim of this randomized, controlled crossover pilot trial was to investigate the influence of isotonic sodium chloride, Ringer and glucose solution on changes in body composition until 120 min post-fluid administration assessed via BIA in healthy individuals. We hypothesized that all solutions due to their osmotically active components impact total body water and thus the calculation of body composition.

## 2. Materials and Methods

This was a single-center, randomized, controlled crossover pilot trial, assessing the impact of different orally administered solutions on body composition measured by bioelectrical impedance analysis (InBody 770, InBody Co., Seoul, Republic of Korea). The local ethics committee of the University of Bayreuth (Germany) approved the study protocol (O 1305/1—GB, 10 June 2021). The study was conducted in conformity with the Declaration of Helsinki and guidelines for Good Clinical Practice. Before any trial-related activities, potential participants were informed about the study protocol and participants gave their written informed consent.

### 2.1. Eligibility Criteria

Eligibility criteria included male or female individuals aged from 18–35 years with a body mass index (BMI) of 18.0–29.9 kg·m^−2^ (both inclusive). This age range was chosen to exclude age-related confounding factors such a diminution of total body water [20] or menopause in women [21]. Individuals were excluded if they were enrolled in a different study, received any pharmaceutical products including investigational medications, or had blood pressure outside of the range of 90–150 mmHg for systolic and 50–95 mmHg for diastolic after resting for five minutes in a supine position. Furthermore, participants were excluded if they suffered from any metabolic disease, including renal, thyroid or liver disease, or had a history of multiple and/or severe allergies to any trial-related products. In addition, females who were pregnant or individuals with a heart pacemaker were also excluded. To assure an euhydrated status prior to the study experiments, participants were also excluded if they demonstrated a urine-specific gravity outside the range of 1005–1030 mg·mL^−1^.

### 2.2. Assessment of Eligibility

Inclusion and exclusion criteria were assessed by an investigator during the screening visit prior to the start of the study.

### 2.3. Study Design

After inclusion in the study, participants were assigned to ascending numbers and then allocated to the order in which the trial visits were conducted following a cross-over randomized fashion set by a software Research Randomizer^®^ (1:1:1:1) [22]. Based on their body mass, participants received an individual quantity of each solution (12 mL per kg body mass), which was measured at the start of each trial visit. Body composition was analyzed before, immediately after, and then every 10 min for 120 min after the fluid consumption using a bioelectrical impedance analysis. Between each trial visit, a minimum of 48 h was maintained, except for the control condition, after which the next visit could take place after a minimum of 24 h.

### 2.4. Trial Visits

Prior to the start of each trial visit, participants had to fast for at least 12 h and refrain from any strenuous physical activity for at least 48 h. Participants were also not allowed to consume alcohol within 24 h before each visit. Upon arrival at the research facility of the University of Bayreuth in the morning after an overnight fast, participants were asked to use the bathroom for bowel emptying, if necessary. A urine specific gravity test was performed to ensure an euhydrated status (Combur^10^, Roche Deutschland Holding GmbH, Grenzach-Whylen, Germany). Subsequently, the participants remained in a seated position for 10 min before a duplicate baseline analysis of body composition was conducted using BIA according to the manufacturer’s specifications (InBody 720, InBody Co., Seoul, Republic of Korea). This body composition analyzer emits a multitude of frequencies from 1 kHz to 1 MHz using the multi-frequency technology which separates the intra- from the extracellular water, thereby minimizing the probability of errors caused by individual variations in the distribution of the total body water or its changes over a given time period.

To determine changes in body composition, the following compartments were analyzed: Intra- (ICW) and extracellular water (ECW), total body water (TBW), skeletal muscle mass (SMM), body fat mass (FM) and visceral fat area (VFA). Following the duplicate baseline measurements (t_rest_), participants received 12 mL per kg body mass of either isotonic sodium chloride (NaCl 0.9%, B. Braun, B. Braun Melsungen AG, Melsungen, Germany), 5% glucose (Glucose-Lösung 5%, Deltamedica, Reutlingen, Germany), or Ringer’s solution (Ringer B. Braun, Melsungen AG, Melsungen, Germany, see Table 1), respectively, which had to be consumed within 60 s. During the control visit, no liquids or similar were consumed. The trial arms are subsequently abbreviated as CON (control), GLU (glucose), ISO (isotonic sodium chloride) and RIN (Ringer). For electrolyte composition and osmolarity of the different solutions, see Table 1. Further BIA was obtained immediately after the fluid consumption (t_0_) and then every 10 min for a total period of 120 min (t_10–120_). Any other form of food and fluid intake was not permitted during each of the trial visits. Between the measurements, the participants remained in a seated position at standardized room temperature (22 °C) and humidity (50%).

### 2.5. Statistical Analysis

All data were assessed for distribution by means of the Shapiro–Wilk normality test. Descriptive statistics are given as mean ± standard deviation (SD), and 95% confidence intervals (CI). Within-group changes were analyzed using a repeated measures ANOVA, Friedman test or mixed-effects model followed by a post hoc Tukey multiple comparisons test. For between-group analyses, a two-way ANOVA or a mixed-effects model was performed. Statistical analyses were conducted using GraphPad Prism Software version 8.0 (GraphPad, La Jolla, CA, USA).

## 3. Results

In total, 18 healthy individuals (10 females) were included with a mean ± SD age of 23.1 ± 1.8 years, height of 176 ± 10 cm, body mass of 69.5 ± 12.5 kg and BMI of 22.2 ± 2.1 kg·m^−2^. All screened participants were eligible to take part in the study, from which no participant had to be excluded; furthermore, no participant had to leave the study prematurely.

Based on the individual body mass on the days of the fluid conditions, participants consumed 831.6 ± 148.9 mL of ISO, 835.3 ± 151.3 mL of GLU and 832.7 ± 149.4 mL of RIN (*p* > 0.99), equaling a ~2% increase in TBW.

Table 2 shows the baseline values for total body water including intra- and extracellular water, skeletal muscle mass, body fat mass and percentage as well as visceral fat area. No significant differences between each condition were found.

### 3.1. Intracellular Water

A two-way ANOVA revealed that there was a statistically significant interaction between the effects of fluid ingestion and time (F [8.51, 135.5] = 3.028, *p* < 0.01). Simple main effects analysis showed that time (*p* < 0.01) but not fluid intake (*p* = 0.90) had a statistically significant effect on changes in intra-cellular water (ICW).

Intracellular water did not show any changes over the 120 min in the CON and GLU trial arms. In the ISO trial arm, ICW demonstrated significant increases from t_rest_ to t_0_ and t_10_, respectively (*p* < 0.01, Figure 1A). In the RIN trial arm, a significant increase between t_rest_ and t_0_ was found (*p* < 0.001).

At t_0_, ICW was significantly increased in both the ISO and RIN trial arms when compared to CON and GLU (*p* < 0.01); however, no changes were found between ISO and RIN (*p* = 0.55). At t_10_, ISO was significantly increased when compared to CON (*p* < 0.05) but not to RIN (<0.99) or GLU (*p* = 0.11).

### 3.2. Extracellular Water

We found a statistically significant interaction between the effects of fluid ingestion and time (F [8.76, 139.6] = 4.688, *p* < 0.0001). Simple main effects analysis showed that time (*p* < 0.01) but not fluid intake (*p* = 0.82) had a statistically significant effect on changes in extra-cellular water (ECW).

ECW did not show any changes in the CON and GLU trial arms, but demonstrated significant increases in the ISO and RIN trial arms throughout the whole time period until t_120_ when compared to t_rest_ (*p* < 0.01, Figure 1B).

From t_0_ to t_10_, only the RIN trial arm was significantly increased compared to CON (*p* < 0.05), but not ISO and GLU. At t_30_, both the RIN (*p* < 0.01) and ISO (*p* < 0.05) trial arms were significantly increased compared to CON, but not GLU with only the RIN trial arm showing significant increases in the further course until t_120_ (*p* < 0.001), while in the ISO trial arm significant increases were also found between t_70_ and t_120_ compared to CON (*p* < 0.01).

### 3.3. Total Body Water

No statistically significant interaction between the effects of fluid ingestion and time (F [1.197, 18.99] = 3.278, *p* < 0.08) were found. Simple main effects analysis showed that neither time (*p* = 0.08) nor fluid intake (*p* = 0.88) had a statistically significant effect on changes in total body water (TBW).

TBW showed no significant differences within the CON and GLU trial arms. For the ISO trial arm, TBW was significantly increased between t_rest_ and t_0_ and t_10_, respectively (*p* < 0.0001). In the RIN trial arm, TBW was significantly increased between t_rest_ and t_0_ (*p* < 0.0001), and between t_0_ and t_10_, t_30_, t_40_, t_50_, t_90_, t_100_ and t_120_ (all *p* < 0.05), respectively.

At t_0_, TBW was significantly increased in the RIN trial arm when compared to CON (*p* < 0.05), but not the other trial arms (*p* < 0.05). At t_10_, both RIN (*p* < 0.05) and ISO (*p* < 0.05) were significantly increased compared to CON, but not GLU (*p* = 0.05 and *p* = 0.25). In the RIN trial arm, TBW was significantly increased until t_120_ when compared to CON (*p* < 0.05).

### 3.4. Skeletal Muscle Mass

We found a statistically significant interaction between the effects of fluid ingestion and time (F [8.390, 133.6] = 2.961, *p* < 0.001). Simple main effects analysis showed that time (*p* < 0.01) but not fluid intake (*p* = 0.90) had a statistically significant effect on changes in skeletal muscle mass (SMM).

There were no significant changes in SMM within the trial arms CON and GLU. In the RIN and ISO trial arms, SMM was significantly increased between t_rest_ and t_10_ (both *p* < 0.01, Figure 1C).

At t_0_ and t_10_, SMM in the RIN and ISO trial arms was significantly increased when compared to CON and GLU (*p* = 0.01 and *p* < 0.05), whereas at t_20_ only the RIN trial arm was significantly higher compared to GLU (*p* < 0.05), but not to CON (*p* = 0.07) and ISO (*p* = 0.77).

### 3.5. Body Fat Mass

With regard to body fat mass (FM), we found a statistically significant interaction between the effects of fluid ingestion and time (F [4.391, 69.80] = 3.517, *p* < 0.01). Simple main effects analysis showed that time (*p* < 0.01) but not fluid intake (*p* = 0.32) had a statistically significant effect on changes in FM.

FM showed no significant changes in the CON and ISO trial arm (Figure 1D). In the RIN trial arm, FM was significantly increased between t_rest_ and t_40_ and t_50_ (*p* < 0.05), and between t_rest_ and t_90_ and t_120_, respectively (*p* < 0.05). In the GLU trial arm, FM was significantly increased between t_rest_ and t_10_ (*p* < 0.0001) and in the further course until t_40_. Additionally, percentage changes in body fat mass can be obtained from Figure 1E.

At t_0_ (*p* < 0.01), t_10_ and t_20_ (*p* < 0.05), FM was significantly higher in the GLU compared to CON and RIN. No significant differences were found between the CON and ISO trial arms.

For visceral fat area (VFA), we found a statistically significant interaction between the effects of fluid ingestion and time (F [9.522, 151.6] = 3.935, *p* = 0.0001). Simple main effects analysis showed that time (*p* < 0.0001) but not fluid intake (*p* = 0.90) had a statistically significant effect on changes in VFA.

For VFA, no significant differences were observed in the CON trial arm. In the ISO trial arm, VFA was significantly increased between t_rest_ and t_20_, t_40_, t_80_ and t_110_, respectively (*p* < 0.05, Figure 1F). In the GLU trial arm, VFA was also significantly increased between t_rest_ and t_0–20_ (*p* < 0.001) and between t_rest_ and t_40_ and t_70_, respectively (*p* < 0.05). In the RIN trial arm, VFA was significantly increased between t_rest_ and t_10_ (*p* < 0.01) and further time points until t_120_.

At t_0_ (*p* < 0.01), t_10_ and t_20_ (*p* < 0.05), VFA was significantly higher in the GLU trial arm compared to CON and RIN. Between the CON and ISO trial arms, and RIN and ISO trial arms, respectively, no significant differences were found.

## 4. Discussion

The aim of this randomized, controlled crossover pilot trial was to investigate the influence of an isotonic sodium chloride, Ringer and glucose solution on changes in body com-position until 120 min post-fluid administration assessed via bioelectrical impedance analysis in healthy individuals. While the CON trial arm demonstrated no significant differences in any of the analyzed body compartments, our results indicated that all fluids independent of their osmolarity exerted a significant influence on body composition with the RIN solution mainly affecting intra- and extracellular water and visceral fat area, while the ISO primarily affected intra- and extracellular water. In contrast, GLU mainly affected body fat mass including visceral fat area.

Previous studies have demonstrated that acute fluid consumption affected measures of impedance and estimated percentage body fat using segmental bioelectrical impedance analysis [14,23,24,25]. Our results are well in line with these investigations, indicating that electrolyte-containing fluids lead to a subsequent increase in impedance and thus TBW and/or body fat mass. At the same time, however, we also found some contradictory results. Gomez et al., for instance, have demonstrated increased measurements of resistance immediately after the consumption of either water, a hypotonic rehydration beverage or an isotonic solution [14]. However, in the isotonic solution trial arm the subsequent changes in resistance had decreased significantly 90 min after the fluid consumption, leading to an overestimation of TBW. This contrasts with our findings where the ISO trial arm did not show any significant difference from baseline measurements except for the initial measurements until t_40_. Moreover, we cannot confirm the postulated underestimation of TBW as a result of the fluid intake when using the BIA. One possible explanation for these conflicting results may be the amount of fluid administered to increase TBW, which was 3% as compared to the 2% in our study. Moreover, we cannot conclusively explain whether other previously mentioned factors such as skin temperature, blood flow or food or fluid intake other than the fluids within the study may have affected the impedance measurements [26,27]. However, because our participants were in a standardized laboratory setting and were only included when they refrained from physical activity and adhered to overnight fasting, these confounding factors should play a minor role when interpreting our results. The fasting itself does not seem to affect total body water as Kuklinski et al. have demonstrated after a 10 h fasting period [28]. It also remains speculative whether changes in estimated TBW are strongest the greater the electrolyte content of the fluids are, as we found greater and longer-lasting effects for the RIN compared to the ISO even though both solutions contain the same amount of osmotically active particles.

Although the ISO and RIN have the same osmolarity (308 vs. 309 mOsm∙L^−1^), the latter has led to greater changes in body composition. This was particularly true for the elicited changes in FM. RIN led to a significant increase in VFA which was detectable directly after the fluid consumption and persisted throughout the entire 2 h period. While it was postulated that rapid increases in impedance about 20 min after the fluid intake are due to the redistribution of blood volume from the periphery to the core in response to a large volume of fluid entering the stomach and gastrointestinal tract [23], the persistent increase until t_120_ could be the result of the intestinal absorption and subsequent dilution of body fluid electrolytes [14]. This mechanism may be a possible explanation for the continuing increase in VFA in the RIN trial arm. However, the same should have been observed at least in the ISO trial arm, since the volume and osmolarity were the same. This led to the conclusion that these changes in VFA must be interpreted with caution. However, due to the electrolyte composition of the ISO and RIN, one might assume that changes in extra- and intracellular water would follow osmotic regulation. Because the ISO is a simple electrolyte solution containing only sodium and chloride, it should first increase the ECW, i.e., blood plasma or interstitial fluids. Osmotic processes should lead to a delayed increase in ICW in order to maintain the ECW/ICW ratio. The RIN, however, also contains potassium and calcium ions in addition to sodium and chloride which would theoretically help to maintain the ECW/ICW ratio without increasing the ECW first. In line with this, the ISO led to a significant increase in the EWC which was detectable immediately after the fluid consumption at t_0_ and persisted until t_120_. However, changes in the ICW were not detected by the BIA except for the time points t_0_ and t_10_ immediately after the fluid intake. This is noteworthy as there are not likely to be any changes in the ICW at these early time points, but rather an increased volume within the core as mentioned earlier.

As far as the GLU is concerned, to the best of our knowledge only a limited number of studies have investigated the effect of a carbohydrate drink on changes in body composition. For instance, Kuklinski et al. found that after the ingestion of a 400 mL carbohydrate drink containing 50.4 g of maltodextrin and fructose (240 mOsm∙L^−1^), no differences in body water distribution or body fat mass were found after the carbohydrate-enriched drink administration [28]. Because the authors performed their measurements 2 h post administration of the carbohydrate drink, these results are generally in line with ours, indicating that carbohydrate-rich fluids do not impair impedance measurements given the fact that enough time has passed between consumption and measurements. Moreover, Dixon et al. investigated whether a 591 mL carbohydrate-electrolyte drink (35 g carbohydrates, 270 mg sodium, 75 mg potassium) affected measures of impedance and percentage body fat estimates using segmental bioelectrical impedance analysis. They reported that 20 min after the fluid consumption, the percentage body fat mass was significantly increased by 1.2%, which is a little higher compared to our results.

While our and previous findings indicated statistical significance, it remains uncertain whether these increases in body fat mass, e.g., between t_rest_ and t_40_ as a result of a carbohydrate-enriched fluid consumption, are of clinical relevance as the mean differences lie below 1%. This is also supported by the fact that in the CON trial arm, values for percentage body fat mass fluctuated by up to 0.7% between t_rest_ and t_120_. This ultimately also applies to the other parameters, as the measured changes triggered by the fluid consumption were rather small and possibly within the error of the impedance technique, as was outlined previously [25]. It was thus concluded that impedance measures, e.g., of body fat mass in clinical settings, do not require strict adherence to a fasting protocol, therefore increasing the opportunities for clinical application. However, they may be of significance in special clinical populations where total body water is of major importance, e.g., in oncological patients or individuals with respiratory, renal or cardiovascular disease. For instance, BIA has repeatedly been used to predict mortality in chronic obstructive pulmonary disease (COPD) patients [29], to estimate body composition in surgical and oncological patients [30] as well as in cancer patients [31]. This was also supported by a systematic review from Haverkort et al., where the authors stated that estimates of body compositions from a BIA can only be useful when performed longitudinally and under the same standard conditions [30], which is strongly supported by our results.

## 5. Conclusions

Our results demonstrated that all fluids may impact conductivity and thus the calculation of body compartments. The RIN and ISO mainly affected increases in extracellular water, while the RIN also affected body fat mass. The GLU solely seems to affect changes in body fat mass including visceral fat area, leading to significant overestimations. We conclude that for the standardized analysis of total body water, skeletal muscle mass and fat mass via a bioelectrical impedance analysis, measurements should take place without any recent fluid intake, while controls are in place for other confounding variables, even though the differences are likely to be clinically insignificant.

## Figures and Tables

**Figure 1 metabolites-13-00473-f001:**
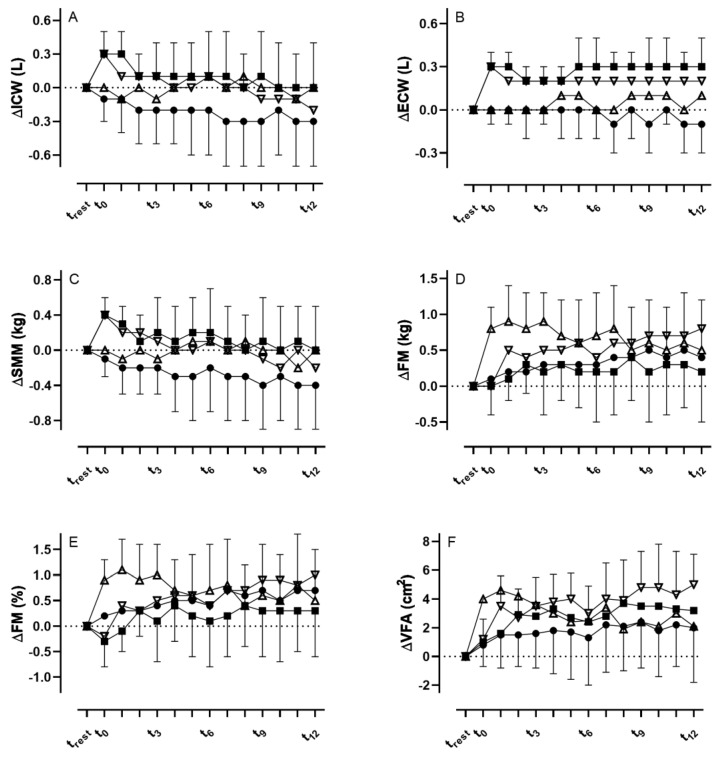
(**A**,**B**) Changes in intra- (ICW) and extracellular water (ECW), (**C**) skeletal muscle mass (SMM), (**D**,**E**), fat mass (FM), and (**F**) visceral fat area (VFA) after the consumption of the three different fluids and during the control visit (• = CON, Δ = GLU, ∇ = RIN, ■ = ISO).

**Table 1 metabolites-13-00473-t001:** Electrolyte composition and osmolarity of the sodium chloride, G5 and Ringer solution.

	0.9% Sodium Chloride	Ringer	G5
Na^+^ (mmol∙L^−1^)	154	147	0
Cl^−^ (mmol∙L^−1^)	154	156	0
Ka^+^ (mmol∙L^−1^)	0	4	0
Ca^2+^ (mmol∙L^−1^)	0	2.2	0
Glucose (g)	0	0	50
Osm ^1^ (mOsm∙L^−1^)	308	309	278

^1^ osmotically active particles.

**Table 2 metabolites-13-00473-t002:** Baseline values (t_rest_) of body composition on the four trial-related visits.

		Mean ± SD	95% CI	*p*-Value
**TBW (L)**	CON	41.3 ± 8.9	36.9–45.8	>0.99
	ISO	41.6 ± 9.0	36.9–46.2	>0.99
	GLU	41.3 ± 9.1	36.7–45.8	>0.99
	RIN	42.1 ± 2.8	37.3–47.0	>0.99
**ICW (L)**	CON	26.0 ± 5.6	23.2–28.8	>0.99
	ISO	26.1 ± 5.6	23.2–29.0	>0.99
	GLU	26.5 ± 5.7	23.4–29.5	>0.99
	RIN	25.9 ± 5.7	23.1–28.8	>0.99
**ECW (L)**	CON	15.4 ± 3.3	13.7–17.0	>0.99
	ISO	15.4 ± 3.4	13.7–17.2	>0.99
	GLU	15.7 ± 3.4	13.8–17.5	>0.99
	RIN	15.4 ± 3.4	13.7–17.0	>0.99
**SMM (kg)**	CON	31.9 ± 7.3	28.3–35.5	>0.99
	ISO	32.1 ± 7.3	28.3–25.9	>0.99
	GLU	32.6 ± 7.5	28.6–36.5	>0.99
	RIN	31.8 ± 7.5	28.1–35.5	>0.99
**FM (kg)**	CON	13.0 ± 3.9	11.0–14.9	>0.99
	ISO	14.7 ± 6.1	11.5–17.8	>0.99
	GLU	12.6 ± 3.7	10.7–14.6	>0.99
	RIN	13.3 ± 3.6	11.5–15.1	>0.99
**FM (%)**	CON	19.0 ± 5.4	16.3–21.7	>0.99
	ISO	19.4 ± 5.1	16.8–22.0	>0.99
	GLU	18.3 ± 5.3	15.5–21.2	>0.99
	RIN	19.4 ± 5.3	16.8–22.1	>0.99
**VFA (cm^2^)**	CON	51.3 ± 19.9	41.4–61.2	>0.99
	ISO	52.8 ± 18.3	43.4–62.2	>0.99
	GLU	49.7 ± 19.2	39.4–59.9	>0.99
	RIN	52.6 ± 18.4	43.5–61.8	>0.99

SD = standard deviation, CI = confidence interval, TBW = total body water, ICW = intracellular water, ECW = extracellular water, SMM = skeletal muscle mass, FM = fat mass, VFA = visceral fat area.

## Data Availability

Data will be made available upon reasonable request by the corresponding author.

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
