# Peer review of "Acute Fluid Intake Impacts Assessment of Body Composition via Bioelectrical Impedance Analysis. A Randomized, Controlled Crossover Pilot Trial"

_metabolites, 2023, doi:10.3390/metabo13040473_

Round 1
Reviewer 1 Report
The study appeared to be well prepared, while corrections for improvement of the paper could be necessary.
1. As the authors stated, the effect of the mild difference of parameters (e.g., body fat) between the groups in clinical settings should be more discussed. Have the effects of such a mild difference in parameters been reported in prior clinical patients’ studies?
2. May the results of the study on younger subjects speculatively be different from those of the older subjects? If it is possible, that may be discussed as a study limitation.
3. Row 61-66; some references can be added to each sentence.
4. Row 74; why did the authors chose the device of InBody?
5. Row 87; were the patients with liver diseases excluded?
6. BIA was abbreviated in row 37, but not in other parts (Row 73 etc.). Unify it.
7. Row 44; the BIA-given results? `-` may be required.
Author Response
Dear Reviewer and esteemed colleague,
first of all, we would like to thank you very much for your time and effort in reviewing this manuscript. We have found the comments to be very helpful in terms of improving the overall quality of the paper. We have taken all comments into account and revised the manuscript accordingly. Enclosed, you may find our reply document.
Kind regards
Janis Schierbauer et al.

Reviewer 2 Report
The work is interesting. It would be an important contribution to science, but in my opinion the major oversights are the small study group, and the large age range of participant recruitment.
I understand that at this stage it would be very difficult to select participants and narrow the range. If the authors succeeded, it would be a significant increase in the value of the work. If not, please justify why the authors decided on such a group and such an age range.
Specific comments
· Introduction - I suggest expanding the introduction to include information on the importance of impedance in medical and scientific diagnostics. For example, electrical impedance tomography (10.1109/embc.2019.8856792 / 10.21037/atm.2017.12.06 ), electrical impedance myography (10.1063/1.5131631 / 10.3390/s22051941 ) , surface electromyographic (10.3390/s22155686 , 10.3390/diagnostics11040580 ).
· L70 - I suggest adding a research hypothesis.
· L82 - 18–65 years - Why did the authors decide on such a large age range ? This age range covers the menopausal period in women. Changes in women's metabolism associated with the mesopause may distort the results obtained.
· L92 – ‘’-1’’ - should be in superscript, and it seems to be in the middle.
· L142 – ‘’ 2.1. Statistical Analysis’’ - suggests add, confidence interval and effect size.
· L151 –‘’ 18 healthy individuals (ten females)’’ -Why did the authors decide to have such a number of respondents. Was it supported by sample size calulation ?
· L161 - Suggests rewording the title and putting descriptions of abbreviations below the table.
· L192 – ‘’ (F [1.197, 18.99 = 3.278, p<0.08)’’ - closing parenthesis missing '']'''
· The references and citation style should be corrected according to the journal's guidelines.
Author Response

(The authors gave the same response as above.)

Round 2
Reviewer 1 Report
The paper was much improved. How could be even the ‘mild’ difference in outcomes between the groups, as observed in the study, be relevant to the clinical settings? – the authors may stress the clinical relevance more greatly. Thank you.
Author Response
Dear Reviewer,
thank you once again for your comment. As to your question "How could be even the ‘mild’ difference in outcomes between the groups, as observed in the study, be relevant to the clinical settings?" we have now integrated further information from our previous reply comment into the manuscript. Please see L. 335 - 342.
We now hope hat the manuscript is to your satisfaction. If you have any furhter comments, please do not hesitate to contact us.
Kind regards
Janis Schierbauer et al.
Reviewer 2 Report
The authors' answers seem acceptable. After reviewing the responses and the revised text, I have the following final comments.
L42 –‘’ (9,10)’’ - this parenthesis should be at the end of the sentence.
L85 - ‘’aged from 18–35 years’’ - If this age range is correct then it is acceptable. The authors can add information as to why they decided on this range. This will improve the justification for conducting the study.
‘’Since this was a pilot study, we did not perform an ‘a priori’ power analysis.’’ - According to the authors' response. If this study is a pilot study please add clear information in the title.
2.1. Statistical Analysis - Effect size has still not been added.
Suggests improving citation and references according to journal guidelines.
Author Response
Dear Reviewer,
we have updated the manuscript according to your specific comments. Please see the attachment.
As to your comment regarding effect size, however, we are not entirely sure what it is you are referring to. If you mean Cohen's d or partial eta squared, Prism does not calcuate effect size when performing a two-way ANOVA. We have therefore reported the corresponding F-value with DFn and DFd as well as the P-value in the results section. This is one of the reasons we included a paragraph in the discussion section that addresses the clinical relevance of our results. If this does not suffice or it is something else entirely that you are referring to, please do not hesitate to contact us again.
Kind regards
Janis Schierbauer et al.

Round 3
Reviewer 2 Report
Currently, the title is correct;
As for statistical calculations - the authors' translation is acceptable.
Despite the authors' assurances, the references and citation style have still not been corrected. However, these are not errors that affect the substantive quality of the work. Cited pages are missing in some items.
I suggest correcting this before publication. I have no substantive comments, the work can be accepted for publication .